# Trends in the National and Regional Transitional Dynamics of Land Cover and Use Changes in Romania

**Alexandru-Ionuţ Petrişor [1],\* [ID], Igor Sirodoev [2] and Ioan Ianoş [3] [ID]**

[1] Doctoral School of Urban Planning, Ion Mincu University of Architecture and Urbanism, 010014 Bucharest, Romania

[2] Igor Sîrodoev, Department of Natural Sciences, Faculty of Natural and Agricultural Sciences, Ovidius University of Constanta, 900470 Constanţa, Romania; igor.sirodoev@univ-ovidius.ro

[3] Interdisciplinary Centre of Advanced Research on Territorial Dynamics, University of Bucharest, 050663 Bucharest, Romania; office@cicadit.ro

\* Correspondence: alexandru.petrisor@uauim.ro or alexandru_petrisor@yahoo.com; Tel.: +40-213-077-191

**Abstract:** The crucial importance of land cover and use changes, components of the 'global changes', for the worldwide sustainable and resilient development results from their negative influence on ecosystem services, biodiversity, and human welfare. Ongoing debates concerning whether the global drivers are more important than the local ones or which are the most prominent driving forces and effects are still ongoing at the global level. In Europe, the patterns of land cover and use changes differ between the west and the east. Property restitution was an important driver of change in Eastern Europe and especially in Romania. This study aimed to look at the land cover and use changes in Romania by their transitional dynamic using Coordination of Information on the Environment (CORINE) data in an attempt to identify long-term spatially and temporally consistent trends. Although generally inconsistent, the results indicate that deforestation and urbanization tend to prevail over other changes, and the development of agriculture slows its pace. Such findings are consequences of unplanned development associated with little environmental awareness. The presence of hotspots where land cover and use changes seem to be clustered can be seen as a feature of ex-socialist countries undergoing economic transition.

**Keywords:** deforestation; agricultural abandonment; urbanization; transition; unplanned development

## 1. Introduction

The three kinds of changes impacting the global environment, namely land cover and use changes, climate changes, and altered global energy flows, make up, altogether, the 'global changes' [1,2]. Such changes result from a multitude of interactions, which facilitate the permanent transfer of matter from one geo-sphere to another, ensuring the progressive increase of the Terra complexity [3]. All these changes have a spatial projection, simply identified in the dynamics of land cover and use. This dynamics affects the social-ecological systems and the nature conservation in the Anthropocene era, expressed by "the accelerating rate of species extinction and global environmental degradation" ([4], p. 137).

This explains why land cover and use changes are at the core of debates concerning sustainability [5], constituting one of the main driving forces of global environmental change [5–11], and justifies the crucial importance of studying them. This process transforms fundamentally natural systems into man-dominated ones [12], modifying the local microclimate and its composition by pollution, modifying the ecosystem functions, equilibrium and resilience, reducing biodiversity through the loss of different natural habitats, and increasing the frequency of infectious diseases [5,13–19]. Consequently,

they have an impact on the level of ecosystem services [20–22], global sustainability and food security [23], determine climate changes through the surface-energy budget and carbon cycle [5,13,24,25], and can change the cultural landscapes [17,26]. Therefore, assessing their impact requires a systemic approach [27,28]. Nevertheless, the land use transitions exhibit multiple, but reversible dynamics [9].

Monitoring land cover and use changes is a challenging process, since the associated costs and the fact that Earth systems change continuously, with or without the intervention of humans, make the continuous monitoring of land impossible. Land is monitored based on snapshots able to pinpoint the changes during consequent evaluations. The drivers of such changes are called, in this article, 'transitional dynamics', defined based on the previous literature as processes consisting of sequences of mechanisms driven only by agent interactions resulting in irreversible changes of land cover and use [29–31]. Other studies [32] used an equivalent term, 'land cover flows'; we prefer 'transitional dynamics' because it is also used in environmental economics and underlines better the link between socioeconomic development and its environmental consequences.

The assessment of land cover and use changes at multiple spatial scales gave birth to a debate; some authors believe that global forces, especially political and institutional changes, are prevalent over the local and regional ones in determining land cover and use changes [9,33–38], while others consider individual decisions at the level of households to be the main drivers [39]. Debates are also present when looking at the driving forces, but also at the effects.

In summary, land cover and use changes can be seen as a consequence of the need for acquiring resources for immediate use—food, shelter and space for daily activities—mediated by institutional factors [5,12,15,40] and can be caused by population growth, depletion of key resources or socio-economic change and innovation, particularly industrialization, which modify the agricultural and forestry practices [7,13,18,41–43]. Additional changes result from them, such as the migration to urban centers [44,45] or dependence of rents by transportation [46]. The need for productive land determines the deforestation in developing countries [47]. In general, cropping, deforestation/forestation and urbanization are considered the most substantial alterations [23,48–50] and their intensity is reflected by the fragmentation of land [51]. Some authors consider that the development of agriculture accounts for most changes [5,11,52]; others believe that although the urban land cover makes up a small share, urbanization generates a disproportionate share of environmental impacts [17,31,53], and is a major driving force of biodiversity loss and biological homogenization [54].

In Europe, land cover and use changes are caused by economic changes and human wellbeing improvement [55–57]. Agricultural restructuring and urbanization are the main driving forces [50,57]. Europe also exhibits differences between the West and the East, between North and South. The increased political stability is associated with fewer and less intensive land cover and use changes in Western Europe compared to Eastern Europe [58]. Cropland abandonment was a result of the institutional and economic shock in Eastern Europe, and of long-term socio-economic transformations such as urbanization and industrialization in Western Europe [59]. The less intensive land exploitation in Eastern Europe in the post-socialist era is confirmed by the growth which is more prominent [58] and concentrated in the metropolitan areas, replacing adjacent land uses (especially agricultural) at a slower rate than in the west [54,60–62]. Secondary changes are reflected by the increase of pastures [58] in the rural areas, which fragments the land, in opposition with the Western part of Europe where large areas are fragmented by long-term urbanization.

In some areas of Western Europe, the abandonment of mountain agricultural land was done due to the need for more fertile plain land [63]; however, the same phenomenon occurs in Latin America, and generally, in developing countries [47], where the need for agricultural land causes deforestation, but marginal agricultural land is abandoned due to urban-rural migration [64]. Cropland abandonment appeared as survival strategies of individual farmers trying to adapt to changing environmental and socio-economic conditions [65] and was favored by the development of road infrastructure [41]. Other studies, relying on in-depth analyses, taking into account the integrated initiatives and the

multi-dimensional ecosystem services, concluded that there is a good correlation between productivity and biodiversity conservation [66,67].

In Eastern Europe, land cover and use changes are a consequence of the conflict of economy and nature [37], but strongly influenced by the post-socialist change of management practices, policies, and strength of institutions [68,69]; therefore, they can be seen as outcomes of adjustments to new social conditions and part of the transition [70]. The abandonment of agricultural land occurred in locations subject to unfavorable natural conditions, where agricultural production was forced in the socialist times by an extensive use of fertilizers and pesticides [60].

Deforestation and cropland abandonment are a consequence of land fragmentation due to post-socialist property restitution [8,11,26,35,43,51,58,60,71–81]; the last is also correlated to the migration of rural population towards the cities [26]. Shrubs colonize abandoned agricultural land [35,60,63]. As a result of agricultural land abandonment and deforestations, post-socialist countries are more vulnerable to climate change [37]. The excessive urban sprawl can be seen as a post-socialist consequence of the state-uncontrolled urbanization [45].

In the particular case of Romania, land cover and use changes were caused by social, economic and political drivers [49,82,83]. Few studies developed at multiple spatial scales (regional and national) [38,84–93], had different goals and focuses and used different methods. However, their common feature was that land cover and use changes were addressed based on their association with specific transitional dynamics. The findings indicated the transitional dynamics specific to Romania, and generally, to transition economies. These are groups of antagonistic phenomena, i.e., deforestation vs. forestation, development vs. abandonment of agriculture, urbanization and other less important drivers, e.g., the construction of dams, draughts, floods, and unidentified changes. While the dynamics of population and its density could be suspected as a potential driver, previous studies have not found any significant correlation between them and some important transitional dynamics, such as urbanization [94]; therefore, more and deeper research is needed.

This research aims to find out if the three periods of CORINE land cover and use data (i.e., 1990–2000, 2000–2006, and 2006–2012) indicate any temporally and spatially consistent trends of the transitional dynamics connected to the Romanian land cover and use changes, by analyzing them in relationship to the socio-economic and political drivers acting during the three periods. In more detail, these trends reflected by the structural changes of land were correlated with the dynamics of land reforms after the collapse of the totalitarian regime and transition to market economy. The post-communist social economic changes frequently have constituted as a strong human pressure on land use. The first decade (1990–2000) has represented an interested period of transition from degradation to chaos and then to stability [86]. The period 2000–2006 was characterized by European reforms, preparing Romania to become EU member, and the last analyzed period (2006–2012) has been marked by a contradictory economic development, i.e., a social-economic boom immediately followed by economic crisis.

## 2. Data and Methods

### 2.1. Data Cllection and Processing

Land cover and use changes were investigated by means of different methods, including the Integrated Spatial Decision Support Systems, in conjunction with economic models [95], Landsat image composites for forest changes [83], or Moderate Resolution Imaging Spectroradiometer (MODIS)-based Normalized Difference Vegetation Index (NDVI) time series used to map abandoned and re-cultivated land [96,97], comparisons of old maps with the current soil maps [10]. However, monitoring such changes without using geospatial technologies and data is not easy [17,21].

This study uses CORINE land cover and use changes data, provided free of charge through the Internet by the European Environment Agency (http://www.eea.europa.eu/data-and-maps/) for the period 1990–2006 and for the period 2006–2012 through the Copernicus Land Monitoring Service (http://land.copernicus.eu/pan-european/corine-land-cover/lcc-2006-2012/view) in a format compatible

with ArcView/ArcGIS (Environmental System Research Institute, Inc., Redlands, CA, USA, 1992). The data projection is ETRS 1989 Lambert Azimuthal Equal Area L52 M10. The re-projection of data unto Stereo 1970 was a mandatory step for using the data and producing the maps [86].

The classification of data used a combination of the schemes utilized in the previous studies [32,84,87–91,93]; the combination was designed to provide a better general overview at the scale of the national territory. Consequently, seven transitional dynamics were defined, as indicated below:

1. Urbanization was defined as the transformation of natural, agricultural, wetland or water areas in urban areas or changes indicating that urban development occurred within the city limits;
2. Forestation: 'forests' are CORINE classes 3.1.1—coniferous forests, 3.1.2—broadleaved forests, and 3.1.3—mixed forests. The term forestation, which conceptually includes afforestation and reforestation [98], is the change of urban, agricultural, wetland or water areas into forests. This definition accounts for the colonization of abandoned agricultural plots by forests [86,99,100], and also for the transformation of the other natural categories into forests;
3. Deforestation is the change of forests into other land use classes;
4. The restructuring of agriculture is the change of different land uses into agricultural land; usually, the transformations of these areas indicate the development of agriculture;
5. The abandonment of agriculture is the process of replacing agricultural areas by non-cultivated areas;
6. Floods are seen as fundamental changes of natural, agricultural or man-dominated systems into wetlands or waters;
7. Others—the term indicates other land use changes, e.g., the construction of dams, outcomes of the draughts or unknown changes, affecting smaller areas.

*2.2. Analysis of Trends and Influence of Human Pressure*

The analysis of trends is based on tabulating the total area for each period and transitional dynamic utilizing the X-Tools extension of ArcView GIS 3.X (Environmental System Research Institute, Inc., Redlands, CA, USA, 1992). Due to the fact that the total area affected by land cover/use changes decreased its size from one period to another, but also due to the different lengths of the three periods (10 years and then 6 years each), several statistical procedures were involved, including (1) the analysis of raw data (in km$^2$), (2) the analysis of ranked data, obtained by ranking all the raw data on the transitional dynamic characterizing each period using the "RANK" function in Excel 2003 to compute the rank of each transitional dynamic among its peers based on the total surface affected, with higher ranks corresponding to higher values, and (3) the analysis of percentile data, obtained by computing the share of the area affected by a given transitional dynamic within the total area affected in a certain period. The need for several types of data (i.e., raw, ranked, and percentile) was the intent to reveal the trends, given some characteristics of the transitional dynamics. For example, since cities make up only a small share of the national territory, urbanization is often masked [90]; for this reason, ranked data are able to diminish the size gaps between changes affecting large areas, such as those related to the agricultural land, and those affecting small areas, such as urbanization. Similarly, percentile data are able to overcome the differences between periods, while preserving the gaps between different transitional dynamics; this was necessary because the total area affected by changes decreased from each period to the next one.

The trends were analyzed computing the Bravais–Pearson coefficient of linear correlation (ρ) between the values of the area affected by a given transitional dynamic in a given period and the length of the period and its corresponding *p*-value using Microsoft Excel 2003. Increasing trends are revealed by negative values of the coefficient.

In order to assess the human pressure, two measures were used, based on public data from the National Institute of Statistics (http://www.insse.ro/cms/en): (1) the change of population between the two endpoints of each period, and (2) the change of density, computed as ratio between the change of

population and the area of each administrative unit. The second measure was used to validate the results, as the size of each administrative unit was a potential confounder in the correlation between the area affected by changes within an administrative unit and its population change; that is, a larger unit could have a larger population and also be subject to changes covering a wider area simply due to the size. Based on these data, correlations were computed for each period and overall for each type of transitional dynamics.

### 2.3. Spatial Analysis

In a nutshell, the spatial analysis used a cluster-based approach to look at the density of occurrences of the same transitional dynamic in a given cell in order to pinpoint processes than cannot be revealed by the raw CORINE data. For instance, forests are not cut entirely on a given parcel (clear cutting); instead, trees are cut off in a patched way from a larger area until the canopy coverage decreases below the threshold needed to classify it as forest [100]; similarly, urbanization occurs by small developments around the large cities [90].

Our approach relied mainly on *spatial cluster analysis* in order to identify changes at the most relevant units at the base spatial reference level. In the next stage, by their aggregation, the dominant types of changes were individualized at the regional scale in order to obtain a better picture of changes at the national level.

The search for the optimum size, based on testing different sizes in an iterative process, started from the idea that the analytical spatial units should be equal in size (a regular grid), and, at the same time, large enough in order to be comparable with the basic Local Administrative Units (LAU 1). Using the European territorial nomenclature, LAU 1 units are identified in Romania with the communes and municipalities [101]. Romania has 3181 LAU 1 units, which cover the national territory of 238,397 km$^2$. This results into an average surface of a single LAU of about 75 km$^2$, or a square grid cell with the side of 8.66 km. That is why we generated a 9-km square grid with 3127 cells covering the entire Romanian territory. The identified changes were aggregated by the grid cells. These were transformed, after the aggregation, into point grids, with the points located in the geometric centers of the grid cells.

We must emphasize that because of this aggregation approach, useful for national scale analyses, our quantitative figures (as well as the maps) presented and discussed further in text reflect changes within a grid cell of 81 km$^2$. For example, if a land plot of 1 km$^2$ changed its use from forest to a different use indicating deforestation, the entire grid cell is marked as affected by deforestation. For this reason, there may be instances in which three different land plots within the same grid cell were deforested during the three different time periods. In such situations, we concluded that the grid cell was affected by changes in all three time periods, even though the actual changes took place in different parts of the cell. Moreover, there may be instances in which different land plots within the same cell undergo opposite processes, such as forestation and deforestation, or development of agriculture and abandonment of agriculture, within the same period. In such cases, we counted the grid cells several times, for each variable separately, i.e., the same cell could be labeled as forestation and deforestation or development of agriculture and abandonment of agriculture at the same time. The number of labels in a cell is equal to the number of processes occurring inside it. Multiple labels are, therefore, allowed. Although this approach makes our analysis coarser and less appropriate for a local scale assessment, it helps us to better pinpoint spatial patterns in data distribution at regional and national scales.

Spatial cluster analysis was made using a *point density-based approach*, implemented via the density-based spatial clustering of applications with noise ('dbscan') package developed for R statistical environment [102]. The advantage of this approach consists in the correct identification of spatial patterns in the distribution of transitional dynamics. The approach requires two main parameters, i.e., (i) minimum number of points that can form a (spatial) cluster, and (ii) how close the points should be to each other in order to be considered as a cluster. Here, the points used in the analysis were the mid-points of the raster cells. The minimal number of points should be greater than the number of dimensions in the analysis. In our case, we used seven independent transitional

dynamics, i.e., forestation, deforestation, development of agriculture, abandonment of agriculture, flood, urbanization, and other changes, and decided that the smallest region should consist of 10 points. The *k-nearest neighbor distances approach* was applied for choosing the distance threshold, which gave us the value of 20,000 (Figure 1d). *These parameters were used in the analysis of all three periods: 1990–2000, 2000–2006, and 2006–2012.*

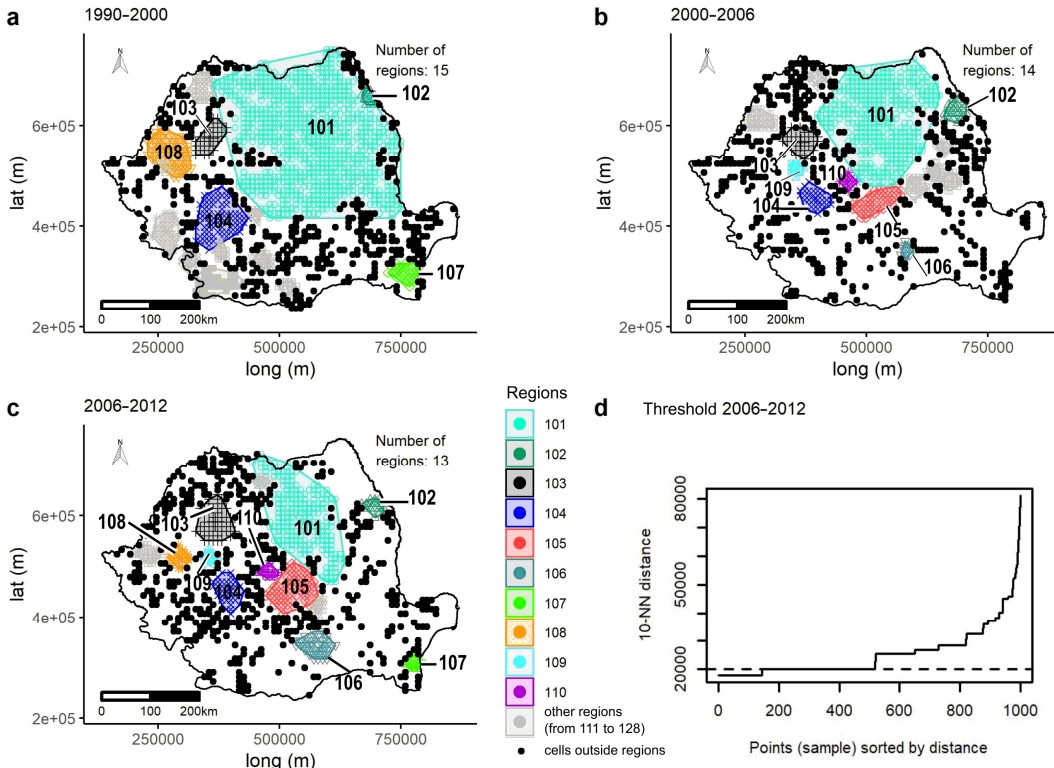

**Figure 1.** Cluster regions of land use changes by three study periods: (**a**) Period 1990–2006; (**b**) Period 2000–2006; (**c**) Period 2006–2012; (**d**) Example of computing the distance threshold for the period 2006–2012. Note: regions shown in colors (numbers from 101 to 110) have a common core in at least two periods; regions shown in grey (numbers from 111 to 128) are characteristic to just one of the study periods.

The spatial clustering procedure was run three times, for each study period independently. The result consists of different number of regions for each period, from 15 (in 1990–2000) to 13 (in 2006–2012) (Figure 1a–c). We also noticed that some regions have common cores in more than one study period, while others are present just in one of the periods. The regions with continuity across all periods change their shape and extent around one stable core. *These regions are of greater interest for our study*. The regions that can be identified in at least two study periods received arbitrary numbers from 101 to 110. Other less important regions, with no continuity, received numbers from 111 to 128.

To explain the spatial changes, we used different statistical information provided by the National Institute for Statistics, and different sources coming from bibliography.

## 3. Results

### 3.1. Statistical Analyses of the National Trends

The analysis of the general trends is represented in Figure 2. The image displays the raw data, the ranked data, and the percentage data.

Each analysis reveals some consistent trends, but they differ from one analysis to another. Raw data reveal decreasing trends for the abandonment of agriculture and deforestation. Ranked data

indicate increasing trends for the development and abandonment of agriculture and decreasing trends for deforestation and urbanization. Percentage data indicate that the abandonment of agriculture slows down, but urbanization increases its intensity.

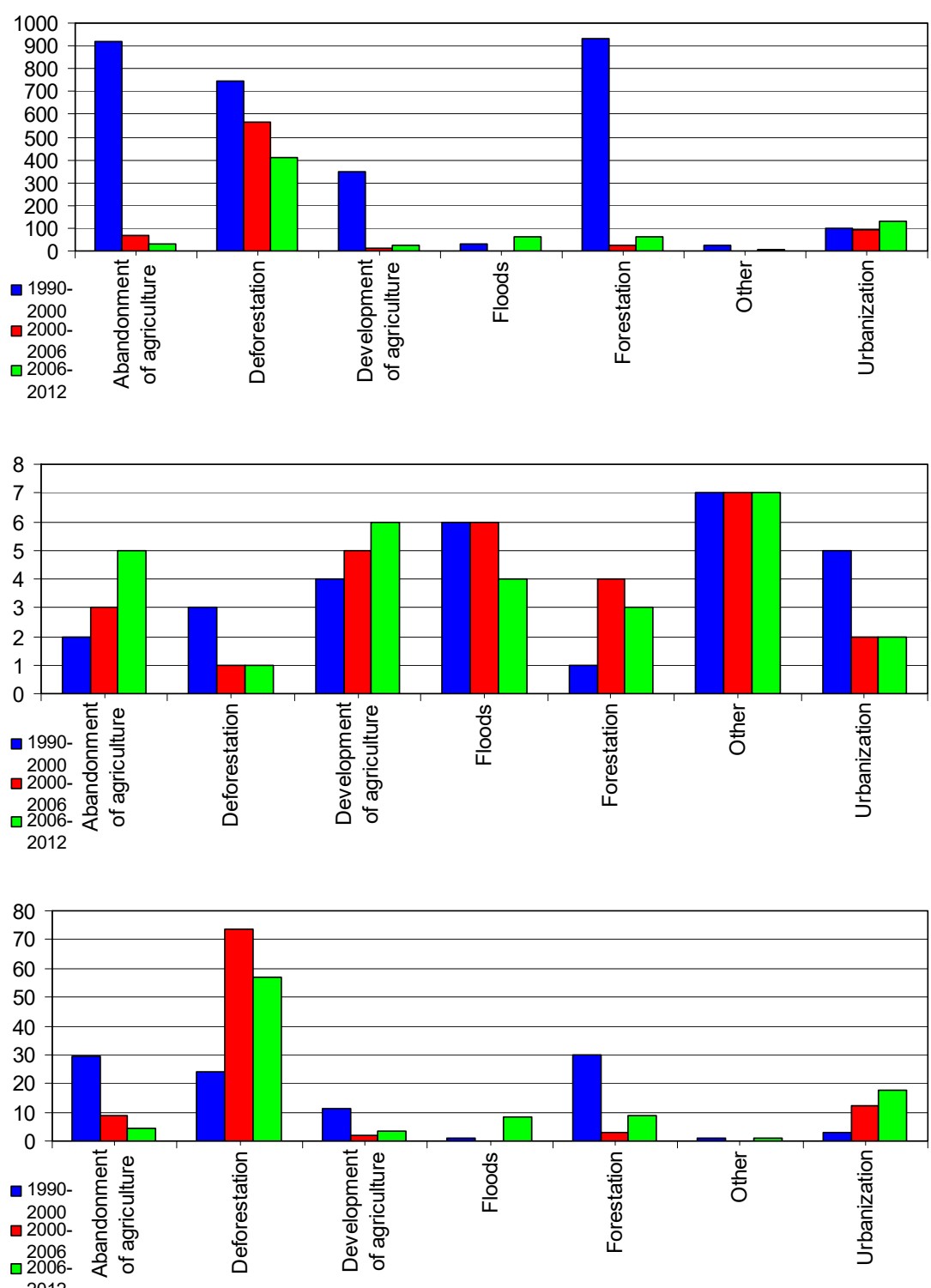

**Figure 2.** Distribution of the total area affected by different transitional dynamics associated to 1990–2012 land cover and use changes in Romania, using CORINE data. The three images reflect the raw data (in km$^2$) (**top**), ranked data (**middle**) and percentage data (**bottom**). The ranking of data was performed using the ascending order.

The results of computing the correlation coefficients and their significance (associated p-values) are showed in Table 1. The results indicate that not all trends are significant. Raw data confirm the decreasing trend of total affected area from 3098.807 km$^2$ in 1990–2000 to 766.365 in 2000–2006 and 719.913 during 2006–2012, also found by previous studies [89,103], and reveal decreasing trends for the abandonment and development of agriculture, and forestation. Ranked data indicate increasing trends for the deforestation and urbanization, while the development of agriculture slows down. Percentage data show no significant trends. These results confirm the previous findings only partially, through the decreasing trend of the development of agriculture.

**Table 1.** Trends of the different transitional dynamics associated to land cover and use changes in Romania during 1990–2012, using CORINE data. The table contains the values of Bravais–Pearson coefficient of linear correlation (ρ) and its associated p-value. ***Bold Italic*** values are significant at α = 0.05. Negative ρ values indicate an increasing trend.

| Transitional Dynamic | Raw Data | | Ranked Data | | Percent Data | |
|---|---|---|---|---|---|---|
| | ρ | *p* | ρ | *p* | ρ | *p* |
| Abandonment of agriculture | *1.00* | *0.02* | 0.76 | 0.45 | 0.99 | 0.11 |
| Deforestation | 0.88 | 0.31 | *−1.00* | *0.00* | −0.94 | 0.22 |
| Development of agriculture | *1.00* | *0.02* | *0.87* | *0.00* | 0.98 | 0.11 |
| Floods | 0.07 | 0.96 | −0.50 | 0.67 | −0.40 | 0.74 |
| Forestation | *1.00* | *0.02* | 0.94 | 0.21 | 0.98 | 0.13 |
| Other | 0.98 | 0.13 | – | – | 0.56 | 0.62 |
| Urbanization | −0.45 | 0.70 | *−1.00* | *0.00* | −0.93 | 0.24 |
| All | *1.00* | *0.01* | – | | | |

## 3.2. Statistical Analyses of the Human Pressure

Table 2 shows the correlations between the population and density change and areas affected by each transitional dynamics for each period and overall. The results indicate the existence of significant correlations between the increase of population and its density and urbanization during each period and overall, with one exception—the correlation between the change of density and urbanization during 2006–2012.

**Table 2.** Correlations between the area affected by each transitional dynamic and population, respectively density change per each Romanian administrative unit within each period covered by CORINE data and overall. The table displays the values of the Bravais–Pearson coefficient of linear correlation (ρ), its associated *p*-value (*p*) and sample size (n). ***Bold Italic*** values are significant at α = 0.05 and **bold** values marginally significant (0.05 < *p* ≤ 0.1). Dark shading indicates statistical significance or marginal significance.

| Transitional Dynamic | Period | Population | | | Density | | |
|---|---|---|---|---|---|---|---|
| | | ρ | *p* | n | ρ | *p* | n |
| **Abandonment of agriculture** | 1990–2000 | 0.00 | 0.89 | 803 | −0.01 | 0.85 | 803 |
| | 2000–2006 | −0.05 | 0.52 | 153 | −0.07 | 0.41 | 153 |
| | 2006–2012 | −0.04 | 0.78 | 59 | −0.02 | 0.88 | 59 |
| | 1990–2012 | −0.01 | 0.81 | 1015 | −0.01 | 0.71 | 1015 |
| **Development of agriculture** | 1990–2000 | 0.02 | 0.68 | 500 | 0.02 | 0.63 | 500 |
| | 2000–2006 | 0.10 | 0.49 | 48 | 0.09 | 0.55 | 48 |
| | 2006–2012 | 0.00 | 0.98 | 32 | −0.09 | 0.62 | 32 |
| | 1990–2012 | 0.02 | 0.69 | 580 | −0.01 | 0.78 | 580 |
| **Deforestation** | 1990–2000 | −0.02 | 0.67 | 621 | 0.00 | 0.96 | 621 |
| | 2000–2006 | 0.03 | 0.38 | 706 | −0.02 | 0.61 | 706 |
| | 2006–2012 | 0.00 | 0.99 | 570 | 0.03 | 0.55 | 570 |
| | 1990–2012 | 0.00 | 0.91 | 1897 | 0.01 | 0.62 | 1897 |

**Table 2.** *Cont.*

| Transitional Dynamic | Period | Population | | | Density | | |
|---|---|---|---|---|---|---|---|
| | | ρ | p | n | ρ | p | n |
| Forestation | 1990–2000 | 0.04 | 0.26 | 811 | 0.05 | 0.13 | 811 |
| | 2000–2006 | *−0.39* | *0.00* | *57* | −0.17 | 0.21 | 57 |
| | 2006–2012 | 0.03 | 0.75 | 106 | 0.06 | 0.57 | 106 |
| | 1990–2012 | 0.03 | 0.34 | 974 | 0.04 | 0.17 | 974 |
| Urbanization | 1990–2000 | *−0.23* | *0.00* | *210* | *−0.19* | *0.01* | *210* |
| | 2000–2006 | *−0.30* | *0.00* | *294* | *−0.15* | *0.01* | *294* |
| | 2006–2012 | *−0.13* | *0.01* | *348* | 0.01 | 0.80 | 348 |
| | 1990–2012 | *−0.18* | *0.00* | *852* | *−0.07* | *0.04* | *852* |
| Floods | 1990–2000 | 0.07 | 0.56 | 65 | −0.01 | 0.96 | 65 |
| | 2000–2006 | −0.89 | 0.11 | 4 | *−0.91* | *0.09* | *4* |
| | 2006–2012 | 0.16 | 0.60 | 13 | 0.19 | 0.53 | 13 |
| | 1990–2012 | 0.04 | 0.73 | 82 | 0.03 | 0.76 | 82 |
| Other unidentified causes | 1990–2000 | −0.17 | 0.19 | 57 | −0.16 | 0.25 | 57 |
| | 2000–2006 | – | – | 0 | – | – | 0 |
| | 2006–2012 | −0.10 | 0.90 | 4 | −0.62 | 0.38 | 4 |
| | 1990–2012 | *−0.23* | *0.07* | *61* | *−0.29* | *0.02* | *61* |

*3.3. Changes by Transitional Dynamic*

Moreover, other correlations were found: a significant one between the increase of population and forestation during 2006–2012, a marginally significant one (0.05 < p ≤ 0.1) between the increase of population and floods during 2006–2012, and another between other unidentified causes and population (marginally significant) and its density (significant) for all periods. We consider that these correlations are spurious, as they appear only during one period, and there are no logical explanations for the relationships.

Table 3 presents a comparison of the changes by the three periods. The first period, 1990–2000, is characterized by the highest proportion of changes in land use types: about 52% of the grid cells were involved in the land changing process. The next two periods, 2000–2006 and 2006–2012, show significant, by 1/3, diminishing of the area affected by change. It is worth mentioning that the difference between the two periods is not high: 34% of land exposed to changes in 2000–2006 and 32% in 2006–2012.

**Table 3.** Cells with changes and with no changes by periods.

| Number of Grid Cells | Periods | | | | | |
|---|---|---|---|---|---|---|
| | *1990–2000* | | *2000–2006* | | *2006–2012* | |
| | Number | % | Number | % | Number | % |
| Total | 3127 | 100 | 3127 | 100 | 3127 | 100 |
| Cells with changes | 1623 | 52 | 1063 | 34 | 998 | 32 |
| No change per period, including | 1504 | 48 | 2064 | 66 | 2129 | 68 |
| *no change in all the periods* | *991* | *32* | *991* | *32* | *991* | *32* |

*3.4. Changes vs. No Changes*

If we look at the changes attributable to each transitional dynamic, we will see that they have different spatial extent. Thus, among all the variables analyzed, *deforestation* is the most significant in terms of the number of cells affected by this process: 1310 cells, which make 41.9% of national territory (Table 4). At the same time, we noticed that deforestation has by far the highest number of cells affected by changes in all the analyzed periods (251). This means that since the beginning of the economic transition period, a significant part of the deforestation process (about 1/5) has been taking

place in more or less the same areas concentrated, especially in Eastern Carpathians (red squares in Figure 3b). Moreover, 625 out of 1310 cells (which represent 48%) were affected by changes in more than one period, which leads us to the idea of a reoccurring deforestation: the process that comes back if not on the same plot of land, then on a neighboring one.

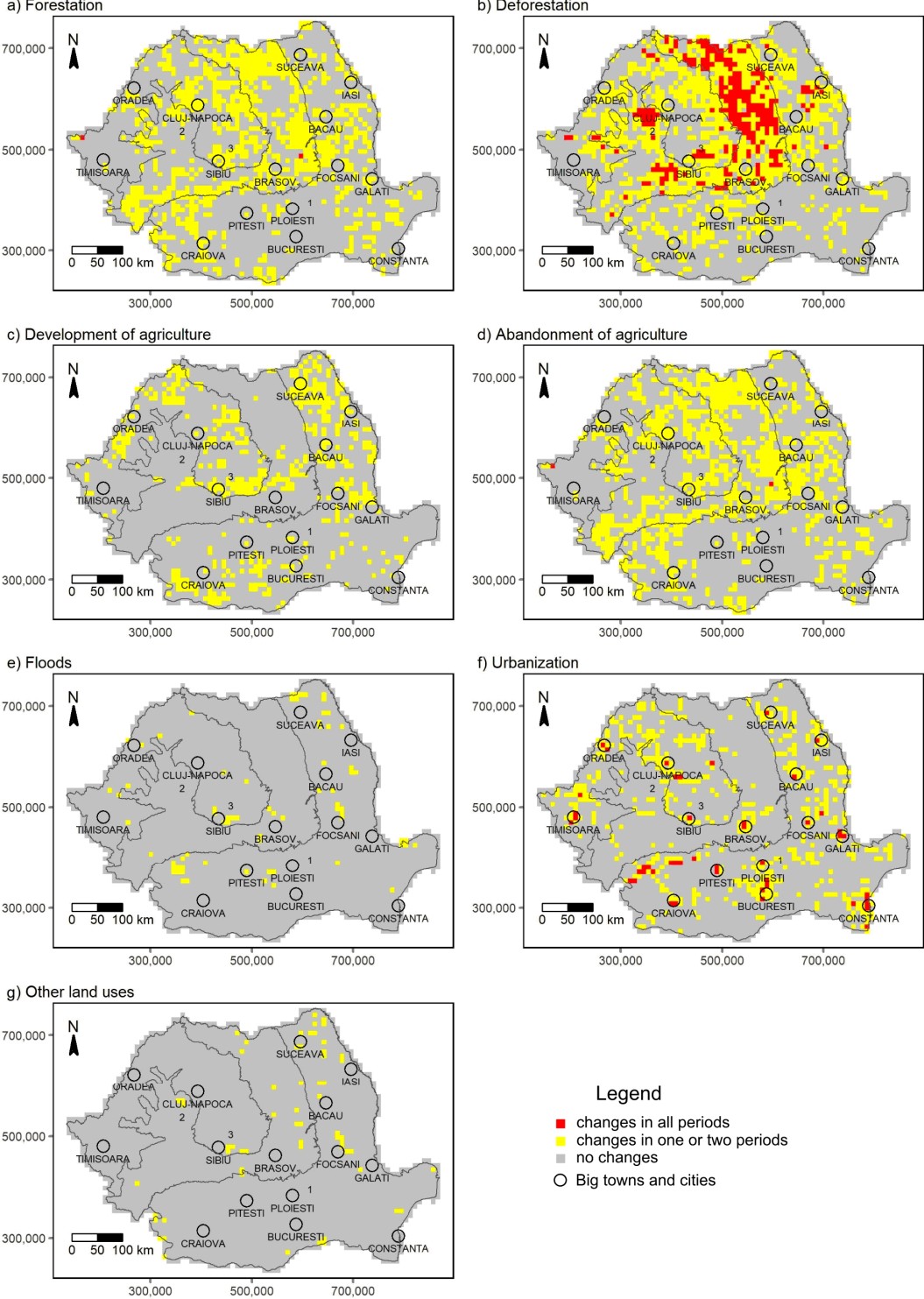

**Figure 3.** Land use changes by grid cells. The image pinpoints areas where LCUC were frequent, occurring in at least one period, and indicating the occurrence of forestation (**a**), deforestation (**b**), development of agriculture (**c**), abandonment of agriculture (**d**), floods (**e**), urbanization (**f**), or other changes (**g**).

**Table 4.** Cells with changes by transitional dynamics.

| Transitional Dynamic | Number of Grid Cells with Changes in: | | | | Share of Cells with Changes in Any of the Periods (%) |
|---|---|---|---|---|---|
| | All Periods | 2 Periods | 1 Period | Any of the Periods | |
| **Forestation** | 2 | 55 | 894 | 951 | 30.4 |
| Deforestation | 251 | 374 | 685 | 1310 | 41.9 |
| Development of agriculture | - | 25 | 473 | 498 | 15.9 |
| Abandonment of agriculture | 2 | 55 | 894 | 951 | 30.4 |
| Flood | - | 4 | 68 | 72 | 2.3 |
| Urbanization | 47 | 116 | 392 | 555 | 17.7 |
| Other uses | - | - | 60 | 60 | 1.9 |

### 3.5. Changes by Regions

Regions were identified based on discontinuities in the density of points with changes in land use. That is why they are not as homogenous, as they are expected to be. At the same time, a higher density of different changes allows for pinpointing areas with less stability in the land use type, regardless of the nature of the drivers causing them, or range of changes. From this viewpoint, ten regions having stable cores with changes occurred in at least two periods reveal the most dynamic areas.

The profiles of the regions are analyzed with respect to the land use changes by grid cells in Figure 4, and with respect to the variation of transitional dynamics by classes and study period in Figure 5. In more detail, Figure 4 allows for looking at the most prominent transitional dynamics affecting each clustering region identified by the spatial analysis during the three periods, while Figure 5 allows for a global comparison of the transitional dynamics across the three periods. The grouping of transitional dynamics is presented using box-and-whisker plots, which are extremely useful for showing variability inside and outside quartiles. Presenting the results in this way is important because the configuration of regions is determined using the rule of vicinity and not that of homogeneity. Such an approach is useful for revealing spatial relationships among regions and their extent. However, the resulting regions are less homogenous, which perfectly justifies the use of box-and-whisker plots.

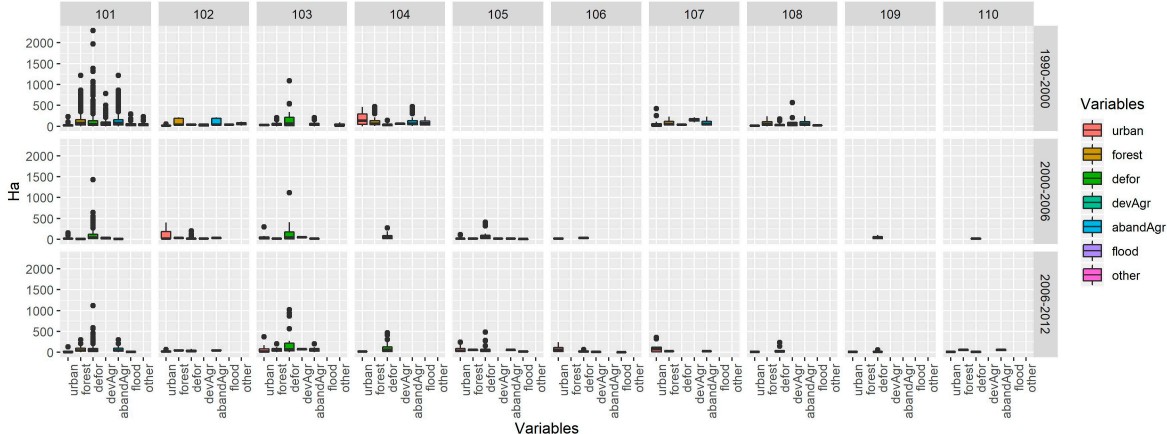

**Figure 4.** Land use changes by grid cells. The variation of transitional dynamics is showed by regions and study period. Transitional dynamics: urban—Urbanization; forest—Forestation; defor—Deforestation; devAgr—Agriculture development; abandAg—abandonment of agricultural; flood—flooded areas; other—other land use changes. Regions as shown in Figure 1.

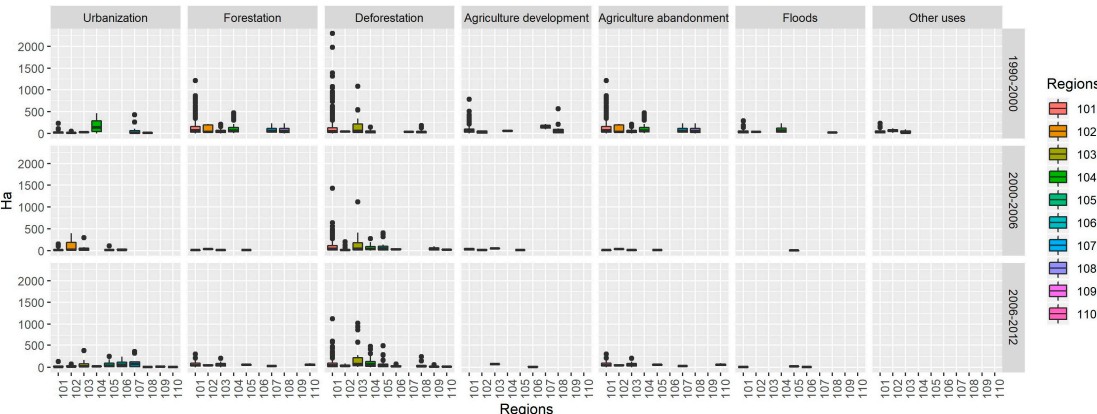

**Figure 5.** Variation of transitional dynamics by classes and study period. Regions as shown in Figure 1.

## 4. Discussions

### 4.1. Statistical Analyses of the National Trends

Overall, the results of the statistical analyses are inconclusive. In addition to the fact that some trends are significant and others are not, the results are not the same when looking at the raw or ranked values and at the percentage-based ones, especially for the transitional dynamics affecting smaller areas; some trends are even reverted (deforestation, floods and urbanization show decreasing trends in terms of raw values, and increasing trends in terms of ranked and percentage data).

The question is which results are the most reliable. Obviously, all are questionable due to the sample size (i.e., correlations based on series of three data), and even the significant correlations may be spurious. However, the raw data simply indicate the area affected by land cover and use changes; these data are not subject to any statistical manipulations and the raw trends, showed in Figure 2 (top) and Table 1, are most likely closer to reality than the others.

### 4.2. Statistical Analyses of the Human Pressure

The analysis of correlations between the population and density change and areas affected by each transitional dynamics for each period and overall is consistent with the previous studies [94].

Resuming the most relevant results, our findings indicate the consequences of unplanned development associated with low environmental awareness. Deforestation and urbanization prevail over the other changes (see the increasing trend), whilst the development of agriculture reduces its pace (see the decreasing trend). Urbanization can be attributed to the demographics, through its correlation with the growth of population and density. The only positive message is that the abandonment of agriculture slows down, also consistent with the previous findings [103]. These findings are not surprising given that the process is connected to the restitution of property [91], which has been already completed. The fact that the total area changed is lesser from one period to another can find an explanation in the different length of periods.

It is also important to say that some results are questionable; for instance, rare events might be subject to spurious correlations or real correlations cannot be detected; this is, for example, the case of floods, with a sample size of only 4.

### 4.3. Changes by Transitional Dynamic

The most visible result obtained is the different extent of changes in correlation with the study period. The first period, 1990–2000, was marked by political, social, economic, and cultural transition in Romania; as a result, it shows the highest proportion of changes in land use types. The next two periods show significant diminishing of the area affected by change.

We should not be surprised by such a huge gap between the last decade of 20th century and the beginning of 21st century. The 1990s were the period in which main land reforms related to transition from centrally planned to market economy, such as returning of state-owned lands to their former, pre-socialist owners were implemented [104,105]. These reforms affected the great majority of agricultural, as well as non-agricultural (e.g., forests [106]) lands, leading to massive changes in their use, revealed by our analysis. The fragmentation degree of agricultural land, for example, was the highest in Europe (about 4,000,000 delimited plots).

At the same time, it is important to mention that 32% of grid cells (one third of national territory) have not been subject to change since the beginning of the transition period 30 years ago. This stability is remarkable in conditions of the double exposure to economic and environmental changes that affected Romania in the past three decades, i.e., land reforms, political transition, economic structuring, 2009–2011 financial crisis and impact of the processes related to global environmental change (desertification, temperature growth, intensification of soil degradation, etc.).

### 4.4. Changes vs. No Changes

*Deforestations* have the highest share among the analyzed changes, with dramatic consequences from the social-economic point of view [107].

The almost perfect correspondence between *forestation* and *abandonment of agriculture* is remarkable. Their similarity in terms of absolute changes (Table 4) and of the spatial pattern (Figure 3a,d) is astonishing but not surprising because forestation usually occurred in Romania on abandoned agricultural lands. Each process has affected about 1/3 of national territory, but the great majority of changes occurred in just one of the periods, with very few repeating patterns (9%, which means 57 cells out of 951).

Unlike previous processes, the *development of agriculture* is less extended, affecting only about 16% of all the grid cells. There are no cells which experienced this process in all three periods. Actually, the great majority of the changes occurred in 1990s [108,109]. In addition, the development of agriculture is the most spatially concentrated process, being confined to large agricultural areas in plains (such as north-western, southern and north-eastern parts of the country) and hill areas in central and eastern parts of Romania (Figure 3c).

*Flooded* areas and *other uses* occupy insignificant parts of the country, about 2% each. Thus, their contribution to the dynamics of land use change at the national scale is very weak. However, at the regional level, floods can represent a significant factor of land use change, such as in Motru River basin [110].

*Urbanization* is a process directly related to the concentration of population [111]. However, the spatial pattern of this process is not well defined except one situation: cells with changes in all three periods (red squares on the Figure 3f) are strongly linked with large cities, such as Suceava, Iaşi, Bacău, Focşani, Galaţi, Constanţa, Bucharest, Ploieşti, Braşov, Cluj-Napoca, Oradea, Arad, Timişoara, Craiova, Piteşti and some other areas, such as those related to mining activities [112].

Compared to other analyzed transitional dynamics, urbanization seems to be a less extended process, covering about 18% of the grid cells. However, it is more important than agriculture development (with 16%) and has more 'reoccurring' situations than forestation and the abandonment of agriculture (47 vs. two cells in all three periods).

An important question is whether there is a connection between urbanization and agriculture. A legal mechanism exists; in order to develop urban areas, parts of the administrative territory changes its legal destination from agricultural to urban, allowing for the expansion of the urban areas over the rural ones. However, only a small share of the agricultural areas is likely to be affected. During the socialist period, by turning small individual plots into large areas owned by the state, agriculture was done extensively. After 1990, the property restitution resulted into the abandonment of agriculture [49,51,52,80,82,97,105,109] and the colonization of abandoned plots by

forest vegetation [86,99,100], but the share of agricultural land taken over by urbanization is not important, since the total urban area is small in Romania [90].

*4.5. Changes by Regions*

The first region, no. 101, is the largest one (over 40% from the national surface). In the 1990s, it covered almost entirely central and north-eastern parts of Romania, which are mainly mountain and hill regions with large forest areas (Figure 4). That is why deforestation here has the highest average and shows extreme figures in all the periods, especially in the last decade of the 20th century [88,89,107,113–115]. Other transitional dynamics, with significant presence in this region, are the forestation and abandonment of agriculture [116,117]. The share of abandoned areas is explained by the high emigration of working population abroad, especially from the rural areas [86,118]. Thus should not be a surprise, because under the pressure of economic transition, agricultural activities started to decline in the areas with more difficult physical conditions. Given the hill and mountain relief of Romania, these areas were re-colonized by forests. This is mainly natural forestation, with very few man-driven forestation initiatives. The extent of this area diminished over the time, overlaying the Eastern Carpathians and Sub-Carpathians.

If the region profiles are compared, region no. 103 is quite similar to the previous one. Its physical characteristics, mountains, and hills, contribute to similar structure and dynamics of land use types. The region has witnessed many transitional dynamics (like no. 101), among which deforestation was the most prominent [108]. At the same time, the region is much smaller and the variability of the transitional dynamics is lower. However, unlike the previous case, this region is extending over time. Moreover, urbanization, playing a secondary role, though, increases its importance here due to development of Cluj-Napoca, one of the most dynamic Romanian cities at present [119]. The effect of economic development could be an extension of its neighborhood in the western part, where the village Floresti has increased its number of inhabitants from 5000 in 1990 to around 40,000 today.

If deforestation is the most prominent transitional dynamic, contributing significantly to the specificity of the first two regions, urbanization is another transitional dynamic with an important impact on several other regions, especially on regions no. 102, 104, 105, 106 and 107, as can be seen in Figure 5.

Region no. 102 is characterized in relation to the land use dynamics around Iaşi. This is why deforestation and urbanization have the biggest impact here, especially in the period that preceded the 2009 economic crisis [109,117,120,121].

Region no. 104 covers an extended mining area of Romania with numerous galleries, sterile rock dumps and open pits. Urbanized areas here are especially characteristic to the first period, due to the fact that mining industrial restructuring in Romania, which has led to closure of numerous mines, enterprises and quarries, opened in the second half of 1990s [122,123]. Since then, deforestation became the leading transitional dynamic in the area.

Region no. 105 became visible after 2000, partly due to the retreat of region 101, but also due to the increasing role of urbanization in its profile, especially after 2006. This region, located in Southern Carpathians, is the most important area for non-sea-side tourism in Romania due to its natural and cultural resources [8,78,91,124]. The main core is formed by Brasov, a city with its surroundings, and Prahova Valley, well known for the mountain resorts. This entire area, being relatively close to Bucharest, the capital city and the largest growth pole in Romania, attracts many tourists and secondary homeowners due to its amenities [125]. These factors explain the increasing role of urbanization in the region's profile and at the same time, the weaker presence of deforestation, which is quite reduced compared to other mountain regions.

Region no. 106 became visible after 2000 and extended significantly after 2006. It is formed around Bucharest. For this reason, urbanization is the leading transitional dynamic in the region's profile [53,61,126,127]. But, surprisingly, in spite of the economic growth of the capital city and its demographic size (six times bigger than the next town in urban hierarchy), this transitional

dynamic does not show its peak here, if compared to other regions. This can be explained by the predominance of brownfield investments in the urban development, which consists of reconfiguring older urban/industrial sites [128–130], rather than by greenfield investment that would have led to the abandonment of agriculture or massive deforestation.

Region no. 107 is the one where urbanization peaks off. The area has a well-developed agriculture and very few forests. At the same time, investments in transport infrastructure and suburbanization of Constanţa, another regional growth pole of Romania, contributed to defining the specificity of this region [61,91,131].

Region no. 108 was found only in the first and the last analyzed periods, when the changes affected the agricultural land, with an important share of abandoned areas [132]. This area is extended at the contact between the Apuseni Mountains and the Romanian Western Plain.

Region no. 109 is located at the contact between the Transylvanian Plateau and Apuseni Mountains covering the urbanized area between Alba Iulia and Aiud towns [133]. The main characteristics are deforestation, and some small built areas surrounding Alba Iulia town.

Region no. 110 is affected by the forestation and deforestation of areas situated in the Tarnave Plateau (part of Transylvanian Plateau), and abandonment of agriculture land, especially surrounding the former industrial area of Copşa Mică (the most polluted town from Romania, until 1990) [134].

Eighteen other regions appear only in one period and are located in different parts of the country. Half of them were individualized in the first period, six in the second, and three in the last one. This means that the regional changes tend to be located in some areas, more and more connected with the urbanization process, on the one hand, and an inverse trend regarding the deforestation which is present in a higher number of regions.

### 4.6. Key Findings

Among the transitional dynamics that influenced land use change in Romania most significantly, two were emphasized: deforestation and urbanization. Deforestation is mainly confined to mountain areas, especially in the Eastern Carpathians. The findings are especially important, because they reveal the 'reoccurring' character of this process: the same areas (but not necessarily same land plots) have been continuously affected by forest cuts since 1990s. Their share is quite high, i.e., 1/5 of all the areas that suffered from deforestation since the beginning of economic transition.

Urbanization is another important process that influences land use change in Romania. Unlike the previous one, it is not spatially concentrated, but confined to large cities, which serve as regional growth poles. Surprisingly, Bucharest, which is 6 times bigger than any other Romanian town, has not witnessed a radically different impact on land use change in its suburban and peri-urban areas. This situation is explained by the fact that after the restructuring of huge industrial enterprises in the urban areas, an important land stock was created. These areas have attracted the developers and investors in other economic sectors such as hotels, business buildings, creative industries, entertainment parks, etc. At the same time, major changes occurred in the immediate sub-urban area, alongside of the main roads, and in its northern part, where the deforestation affected some hectares.

Deforestation and urbanization were the main two drivers of land cover and use changes. They acted mostly in several regions with cores more or less stable over time. This is why these regions either correspond to areas covered by forests in mountains and hills or relate to the large cities with active suburbanization processes. Their size and intensity of land use change have diminished over time. However, some of them, close to the economically dynamic regional growth poles (such as Iaşi, Cluj-Napoca or Bucharest) and driven by urbanization, are still increasing their coverage.

### 4.7. Methodological Limitations and Future Research Directions

The study is subject to general limitations characteristic to the use of CORINE data, including misclassification, and different resolutions and classification schemes from one period to another [85,86,135–137]. In addition to them, the classification of transitional dynamics can

change the results. There are studies focused on different issues, such as the dynamics of forests (forestation vs. deforestation) [10,42,49,74,76,79,81,83,93,98], agriculture (agricultural development vs. abandonment) [49,51,52,80,82,97,105,109], or urbanization [61,62,69,85,90,94,118,121,126,130]; in these cases, the methodologies were fine-tuned in order to discern specific processes. However, in the present study the methodology was adapted to reflect the most important transitional dynamics, identified by the previous studies carried out in Romania, in order to provide a global image, consistent with the intent of identifying long-term trends.

At the time when the present study was carried out, the dataset including the 2012–2018 land cover and use changes was not available. The timing associated with the publication of the manuscript with the Special Issue of "Remote Sensing" on "CORINE Land Cover System: Limits and Challenges for Territorial Studies and Planning" did not allow for redoing the analysis in order to provide an up-to-date image. However, simple assessments of the raw data indicated that the changes occurred during 2012–2018 reveal similar or continuing trends; the total area affected by changes is smaller than the one during 1990–2000, and the dominant transitional dynamics are deforestations, agricultural abandonment, and urbanization. Therefore, the inclusion of the newer dataset is not likely to affect significantly the most important findings of this study or diminish their scientific value.

In this study, different grid sizes were tested, i.e., $5 \times 5$ km, $9 \times 9$ km, and $25 \times 25$ km; this was a methodological study by itself. The results indicate that the optimal size corresponded to the average size of LAU 1 units. However, the detection of land cover/land use changes is significantly influenced by the spatial scale used in the study. The results are a consequence of the fixed grid size of $9 \times 9$ km. The choice of another grid size could influence the results. Nevertheless, the use of a grid approach for analyzing CORINE data is efficient in terms of processing and illustration, but the selection of a proper grid size requires more and deeper research.

Last but not least, although other data sets (e.g., High Resolution layers) and software—e.g., QGIS (Open Source Geospatial Foundation, Chicago, IL, USA, 2007), ArcGIS (Environmental System Research Institute, Inc., Redlands, CA, USA, 1999) etc.—are available, the methodology is commensurate to the skills, possibilities and availability of data at the moment when the research was done. Future research can use a fine-tuned approach involving all the above (i.e., changes of the classification schemes, data, or software) to pinpoint details that might have escaped from the current study due to these methodological limitations.

## 5. Conclusions

This study aimed to look at the most important transitional dynamics characterizing the transition from a totalitarian regime to a liberal open-market economy. The findings are important from this perspective not only for Romania, but for other countries that have undergone similar processes.

In a nutshell, the greatest changes in land use occurred in Romania in 1990s. It was the period of massive change in land ownership from state-owned to private, significant restructuring of economic activities with the decline of those based on extensive type of production (socialist agriculture, mining and quarrying, heavy industry), as well as the period in which local communities tried to re-launch economic growth by selling natural resources, such as wood. These efforts and trends had led to the situation in which by the end of the period, about half of the country (half of the grid cells covering national territory) was affected by land changes to a certain extent.

The method used to analyze one of the most dramatic transitions from a strongly centralized country to a democratic regime and an open-market economy has a high extrapolation power to be used in other countries exhibiting similar development patterns. The results can be used by national and regional decision-makers to define appropriate policies for mitigating the social-economic impact over land cover and use of the processes which develop in conjunction with the collapse of totalitarian regimes, especially at the local level.

Further research can improve this methodology, especially by addressing, at a deeper level, the correlation between land cover and use changes and the social and economic transformations,

and also by defining possible tools for a better understanding of territorial dynamics. Our findings demonstrate the importance of using databases created at the continental scale in order to pinpoint the land use changes and facilitate comparative analyses between regions characterized by different development stages.

**Author Contributions:** All the authors have equally contributed to the article. In more detail, the contributions for the different sections are: conceptualization: A.-I.P. and I.I.; methodology, A.-I.P., I.S. and I.I.; formal analysis, A.-I.P., I.S. and I.I.; investigation, I.I., A.-I.P., I.S.; project administration, A.-I.P.; writing—original draft preparation, A.-I.P., I.I. and I.S. All authors have read and agreed to the published version of the manuscript.

**Funding:** The article was partly supported by the University of Bucharest project UB-2008 "*Trans-scale analysis of the territorial impact of current climate change and globalization*".

**Conflicts of Interest:** The authors declare no conflict of interest. The funders had no role in the design of the study; in the collection, analyses, or interpretation of data; in the writing of the manuscript, or in the decision to publish the results.

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
