# Peer review of "Trends in the National and Regional Transitional Dynamics of Land Cover and Use Changes in Romania"

_remotesensing, doi:10.3390/rs12020230_

Round 1

Reviewer 1 Report

General remarks:

The term " transitional dynamics"  is not explained/introduced in the text and no references are found in the text to relevant literature using/explaining this terminology. In other literature (e.g. Feranec et al, 2010) and studies of the EEA the term land cover flows are used and well described. No reference to this studies are taken into account and used in the definitions of forestation , abandonment etc....

Introduction:

not to the point and relatively long introduction a relative limited objective of "indicating trends in land cover/use for the period 1990-2012" and "interpretation in relation to Romanian situation". No link to analysis/correlation with population data last section 109-199 : last part should be discussions/conclusions

Data methods:

7 land use change categories are introduced  and named/defined as transitional dynamics..... which is confusing. No reference to already existing studies on grouping of land cover/use changes would be good to have something in the discussion or introduction on previous studies and why deviating or propose new categorization. line 134: classification of data...….which classification and/or data the statistical procedures mentioned in line 158-160 should be better described /explained and what are the differences one method is called row data (line 158, 242...) whereas in line 235, 248 it is called raw data, but more important what are the differences between the statistical procedures and why are they applied..... line 176: basic level ? line 205-224;: figure 1 is discussed/analysed which should not be in the data/method section.

Results:

Figure 2: missing (area in ha/km2?); not clear what are the different statistical procedures shown Table 1: Heading  introduces 3 methods with different names as introduced in section 2. Area, rank and % should be explained. Furthermore I see a duplication of abandonment of agriculture...…. line 249 I do not see this trend  reflected in the Figure 1 Format Table 2 should be adapted with lines between the different transitional dynamics section 3.3 is not clear. Interpretation of results should be in discussion. " Changes by transitional dynamics" what is meant with this terminology line 288/289: land use changes caused by deforestation... what do you want to say....deforestation is a land use change? Table 4: For my understanding and it should be better explained....cells are labelled as  one of the 7 transition dynamics (e.g. deforestation). Are only changes in the grid cell taken into account that are considered as deforestation (so no other type of changes). If a grid cell is labelled as deforestation it cannot be labelled as urban development? How the girds are labelled with transitions...on basis of majority? Figure 4 and 5 are not clear! Should be explained!

Author Response

The response is in the attached file; the way of addressing each comment is marked using a blue font.

Reviewer 2 Report

I find the aims of this study very interesting, but I have some major concerns about how the CORINE land cover dataset is used in the analyses making the results biased.

The latest CORINE land cover dataset from 2018 is not included in the study. The dataset is freely available for downloading her: https://land.copernicus.eu/pan-european/corine-land-cover/clc2018 and Romania is included. Not including the latest update of the time series makes the manuscript not up to date, and the newest trends about land cover changes in Romania is not showed or discussed. The CORINE land cover dataset itself is NOT intended to be used for analyzing land cover changes. For analyzing land cover changes between neighboring survey years CLC-changes should be used. This is informed at the Copernicus home page https://land.copernicus.eu/pan-european/corine-land-cover «If you are interested in CLC-Changes between two neighbor surveys always use the CLC-change layer.” Using the CLC dataset for changes analysis will result in missing changes due to scale, and false changes due to the generalization rules used in the production of the CLC datasets. See https://land.copernicus.eu/user-corner/technical-library/copy_of_CLC2006_technical_guidelines.pdf p. 65 for the generalization matrix of the CLC-changes layer for updating of the CLC datasets between reference years. The same matrix has been used for all the CLC updates. CLC-change databases are available for Romania for all the reference years of CLC. In the reclassification of the CLC nomenclature (line 124 ->) the CLC class 243 “Land principally occupied by agriculture, with significant areas of natural vegetation” is reclassified to “Agriculture”. This mixed class are occupied by 25 – 75 % natural- or semi natural vegetation (see https://land.copernicus.eu/user-corner/technical-library/corine-land-cover-nomenclature-guidelines/docs/pdf/CLC2018_Nomenclature_illustrated_guide_20190510.pdf p. 62). Handling this class solely as agriculture areas will bias the estimation of the total amount of agricultural land and abandonment of agriculture inside this class is missed out.

I recognize the “grid approach” as a good solution for analyzing the spatial distribution of the land use changes.

For further analyses of land use changes in Romania other pan-European components such as High Resolution layers, and local components such as Urban Atlas should be considered. All of them available for downloading through the Copernicus home page for several reference years.

Have the authors considered using the free software QGIS (https://qgis.org/en/site/) for the GIS analysis? I see the software ArcView GIS 3.X is mentioned in the text (line 155). This is a very old and outdated software from ESRI. I have no connections or other interest in QGIS it is just a suggestion.

Author Response

(The authors gave the same response as above.)

Round 2

Reviewer 2 Report

This is my second time reviewing this manuscript. My three main comments are:

I still mean that the CHA2018 should be included in the study, to make the manuscript “up to date”. I can not see that the inclusion of the dataset is especially time consuming. The table of the ranked dataset is hard to interpret. I am not sure if the interpretation is correctly done. The authors should revise the data and/or specify. Table 4 and 5 is too small to be readable and I could not review them.

My other comments to the manuscript as follows:

Line 49-57: This is an important point. Very good inclusion.

Line 170: specify the denomination used for the raw data.

Line 170-173: the wording is a bit unclear. Is the rank of the different classes based on the raw data or the percentage? Is the largest change class marked with the largest or smallest number?

Line 203-249: Reorganize the two paragraphs. All the content is relevant for the analysis conducted, but it is a bit hard to follow due to the organization of the two paragraphs.

Line 206-212: I would have chosen a smaller grid size for the analysis based on personal experience with corresponding analysis with national data (Larger country than Romania), but I acknowledge the authors argumentation for a corresponding grid size as the LAU 1.

Line 231-241: Please specify that the points used in the cluster analysis are the mid-point of the raster cells.

Line 260: row instead of raw. Same misspelling several other times.

Line 270: reference to the studies.

Line 276: I find the bar graph for the rank data hard to interpret without some specifications. Are the “top” class given the highest or the lowest number? Intuitively I interpret the “top” class for each period to have the lowest number, but then the interpretation given by the authors in line 263-265 is not in accordance with the seen results in the graph. I think that the bar graph for the ranked data should have been interpreted the “opposite” way; an increase in number is actually a decrease in observed changes for the given class. Then the results seen in the rank analysis gives corresponding results with the two other analyses preformed (line 262-266).

Line 289-292: Table 1: the interpretation of the ranked data should be reviewed by the authors. Should the heading be “percent” and not “percentile”?

Line 296: the sample size of floods are very small n = 4, should be pinpointed.

Line 348-353: Different grid sizes should be tested. It is probably a study in itself. Detection of land cover/land use changes is highly influenced by the chosen scale. The results are most likely affected by the fixed grid size of 9x9 km. Using another grid size would influence the results (personal experience). Using a grid approach for analyzing LULC data is efficient, both for processing and illustration, but the selection of a proper grid size using the approach is an immature field. This should also be discussed in ch 4.

Line 362-370: Figure 4 and figure 5 is too small to be readable. I’m not able to review the results.

Line 384-387: I mean that it is important for this study to include the CHA2018 dataset to strengthen the trend seen for the two last periods.

Line 420-424: Does the urbanization affect the agricultural areas? If Romania has a large potential for high quality agricultural areas, this is probably not relevant. But if the agricultural areas with the highest quality are situated around the big cities, where the urbanization takes place, I think this is relevant to discuss.

Line 430-499: I like this part and find it interesting. Good discussion.

Line 511-520. I can not recognize that the inclusion of another dataset is very demanding for this special study, since most of the data processing and analyzes are done separately for each period. The upcoming of the CLC2018/CHA2018 should be known by the authors when the study was conducted and should have been included at once when the dataset were made available.

Line 528-574: The conclusion is very long. Please take out the most important findings of the study and put the rest in the discussion part.

Author Response

Thank you for the additional comments and for the time and efforts invested in reviewing our submission. Our response is in the attached file.

This manuscript is a resubmission of an earlier submission. The following is a list of the peer review reports and author responses from that submission.